# When Are Overcomplete Topic Models Identifiable? Uniqueness of Tensor Tucker Decompositions with Structured Sparsity

**Animashree Anandkumar**
University of California
Irvine, CA
a.anandkumar@uci.edu

**Daniel Hsu**
Columbia University
New York, NY
djhsu@cs.columbia.edu

**Majid Janzamin**
University of California
Irvine, CA
mjanzami@uci.edu

**Sham Kakade**
Microsoft Research
Cambridge, MA
skakade@microsoft.com

## Abstract

Overcomplete latent representations have been very popular for unsupervised feature learning in recent years. In this paper, we specify which overcomplete models can be identified given observable moments of a certain order. We consider probabilistic admixture or topic models in the overcomplete regime, where the number of latent topics can greatly exceed the size of the observed word vocabulary. While general overcomplete topic models are not identifiable, we establish *generic* identifiability under a constraint, referred to as *topic persistence*. Our sufficient conditions for identifiability involve a novel set of "higher order" expansion conditions on the *topic-word matrix* or the *population structure* of the model. This set of higher-order expansion conditions allow for overcomplete models, and require the existence of a perfect matching from latent topics to higher order observed words. We establish that random structured topic models are identifiable w.h.p. in the overcomplete regime. Our identifiability results allow for general (non-degenerate) distributions for modeling the topic proportions, and thus, we can handle arbitrarily correlated topics in our framework. Our identifiability results imply uniqueness of a class of tensor decompositions with structured sparsity which is contained in the class of *Tucker* decompositions, but is more general than the *Candecomp/Parafac* (CP) decomposition.

**Keywords:** Overcomplete representation, admixture models, generic identifiability, tensor decomposition.

## 1 Introduction

A probabilistic framework for incorporating features posits latent or hidden variables that can provide a good explanation to the observed data. Overcomplete probabilistic models can incorporate a much larger number of latent variables compared to the observed dimensionality. In this paper, we characterize the conditions under which overcomplete latent variable models can be identified from their observed moments.

For any parametric statistical model, identifiability is a fundamental question of whether the model parameters can be uniquely recovered given the observed statistics. Identifiability is crucial in a number of applications where the latent variables are the quantities of interest, e.g. inferring diseases

(latent variables) through symptoms (observations), inferring communities (latent variables) via the interactions among the actors in a social network (observations), and so on. Moreover, identifiability can be relevant even in predictive settings, where feature learning is employed for some higher level task such as classification. For instance, non-identifiability can lead to the presence of non-isolated local optima for optimization-based learning methods, and this can affect their convergence properties, e.g. see [1].

In this paper, we characterize identifiability for a popular class of latent variable models, known as the *admixture* or *topic* models [2, 3]. These are hierarchical mixture models, which incorporate the presence of multiple latent states (i.e. topics) in documents consisting of a tuple of observed variables (i.e. words). In this paper, we characterize conditions under which the topic models are identified through their observed moments in the overcomplete regime. To this end, we introduce an additional constraint on the model, referred to as *topic persistence*. Intuitively, this captures the "locality" effect among the observed words, and goes beyond the usual "bag-of-words" or *exchangeable* topic models. Such local dependencies among observations abound in applications such as text, images and speech, and can lead to more faithful representation. In addition, we establish that the presence of topic persistence is central to obtaining model identifiability in the overcomplete regime, and we provide an in-depth analysis of this phenomenon in this paper.

## 1.1 Summary of Results

In this paper, we provide conditions for *generic*[1] model identifiability of overcomplete topic models given observable moments of a certain order (i.e., a certain number of words in each document). We introduce a novel constraint, referred to as *topic persistence*, and analyze its effect on identifiability. We establish identifiability in the presence of a novel combinatorial object, named as *perfect $n$-gram matching*, in the bipartite graph from topics to words (observed variables). Finally, we prove that random models satisfy these criteria, and are thus identifiable in the overcomplete regime.

**Persistent Topic Model:** We first introduce the $n$-persistent topic model, where the parameter $n$ determines the so-called persistence level of a common topic in a sequence of $n$ successive words, as seen in Figure 1. The $n$-persistent model reduces to the popular "bag-of-words" model, when $n = 1$, and to the single topic model (i.e. only one topic in each document) when $n \to \infty$. Intuitively, topic persistence aids identifiability since we have multiple views of the common hidden topic generating a sequence of successive words. We establish that the bag-of-words model (with $n = 1$) is too non-informative about the topics to be identifiable in the overcomplete regime. On the other hand, $n$-persistent overcomplete topic models with $n \geq 2$ are *generically* identifiable, and we provide a set of transparent conditions for identifiability.

**Deterministic Conditions for Identifiability:** Our sufficient conditions for identifiability are in the form of expansion conditions from the latent topic space to the observed word space. In the overcomplete regime, there are more topics than words, and thus it is impossible to have expansion from topics to words. Instead, we impose a novel expansion constraint from topics to "higher order" words, which allows us to handle overcomplete models. We establish that this condition translates to the presence of a novel combinatorial object, referred to as *perfect $n$-gram matching*, on the bipartite graph from topics to words, which encodes the sparsity pattern of the topic-word matrix. Intuitively, this condition implies "diversity" of the word support for different topics which leads to identifiability. In addition, we present tradeoffs between the topic and word space dimensionality, topic persistence level, the order of the observed moments at hand, the maximum degree of any

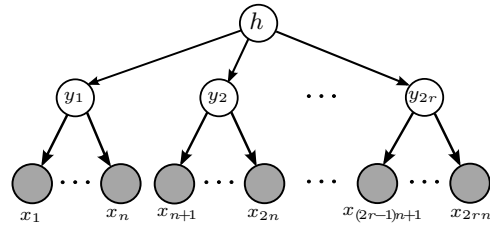

Figure 1: Hierarchical structure of the $n$-persistent topic model. $2rn$ number of words (views) are shown for some integer $r \geq 1$. A single topic $y_j, j \in [2r]$, is chosen for each $n$ successive views $\{x_{(j-1)n+1}, \ldots, x_{(j-1)n+n}\}$.

---
[1]A model is generically identifiable, if all the parameters in the parameter space are identifiable, almost surely. Refer to Definition 1 for more discussion.

topic in the bipartite graph, and the *Kruskal rank* [4] of the topic-word matrix, for identifiability to hold. We also provide the discussion that how $\ell_1$-based optimization program can recover the model under additional constraints.

**Identifiability of Random Structured Topic Models:** We explicitly characterize the regime of identifiability for the random setting, where each topic $i$ is randomly supported on a set of $d_i$ words, i.e. the bipartite graph is a random graph. For this random model with $q$ topics, $p$-dimensional word vocabulary, and topic persistence level $n$, when $q = O(p^n)$ and $\Theta(\log p) \le d_i \le \Theta(p^{1/n})$, for all topics $i$, the topic-word matrix is identifiable from $2n^{\text{th}}$ order observed moments with high probability. Furthermore, we establish that the size condition $q = O(p^n)$ for identifiability is tight.

**Implications on Uniqueness of Overcomplete Tucker and CP Tensor Decompositions:** We establish that identifiability of an overcomplete topic model is equivalent to uniqueness of the observed moment tensor (of a certain order) decomposition. Our identifiability results for persistent topic models imply uniqueness of a structured class of tensor decompositions, which is contained in the class of *Tucker* decompositions, but is more general than the *candecomp/parafac* (CP) decomposition [5]. This sub-class of Tucker decompositions involves structured sparsity and symmetry constraints on the *core tensor*, and sparsity constraints on the *inverse factors* of the decomposition.

Detailed discussion on overview of techniques and related works are provided in the long version [12].

## 2   Model

**Notations:** The set $\{1, 2, \ldots, n\}$ is denoted by $[n] := \{1, 2, \ldots, n\}$. The cardinality of set $S$ is denoted by $|S|$. For any vector $u$ (or matrix $U$), the support denoted by $\mathrm{Supp}(u)$ corresponds to the location of its non-zero entries. For a vector $u \in \mathbb{R}^q$, $\mathrm{Diag}(u) \in \mathbb{R}^{q \times q}$ is the diagonal matrix with $u$ on its main diagonal. The column space of a matrix $A$ is denoted by $\mathrm{Col}(A)$. Operators "$\otimes$" and "$\odot$" refer to Kronecker and Khatri-Rao products [6], respectively.

### 2.1   Persistent topic model

An admixture model specifies a $q$-dimensional vector of topic proportions $h \in \Delta^{q-1} := \{u \in \mathbb{R}^q : u_i \ge 0, \sum_{i=1}^q u_i = 1\}$ which generates the observed variables $x_l \in \mathbb{R}^p$ through vectors $a_1, \ldots, a_q \in \mathbb{R}^p$. This collection of vectors $a_i, i \in [q]$, is referred to as the *population structure* or *topic-word matrix* [7]. For instance, $a_i$ represents the conditional distribution of words given topic $i$. The latent variable $h$ is a $q$ dimensional random vector $h := [h_1, \ldots, h_q]^\top$ known as proportion vector. A prior distribution $P(h)$ over the probability simplex $\Delta^{q-1}$ characterizes the prior joint distribution over the latent variables (topics) $h_i, \ i \in [q]$.

The $n$-persistent topic model has a three-level multi-view hierarchy in Figure 1. In this model, a common hidden topic is persistent for a sequence of $n$ words $\{x_{(j-1)n+1}, \ldots, x_{(j-1)n+n}\}, j \in [2r]$. Note that the random observed variables (words) are exchangeable within groups of size $n$, where $n$ is the persistence level, but are not globally exchangeable.

We now describe a linear representation for the $n$-persistent topic model, on lines of [9], but with extensions to incorporate persistence. Each random variable $y_j, j \in [2r]$, is a discrete-valued $q$-dimensional random variable encoded by the basis vectors $e_i, \ i \in [q]$, where $e_i$ is the $i$-th basis vector in $\mathbb{R}^q$ with the $i$-th entry equal to 1 and all the others equal to zero. When $y_j = e_i \in \mathbb{R}^q$, then the topic of $j$-th group of words is $i$. Given proportion vector $h \in \mathbb{R}^q$, topics $y_j, j \in [2r]$, are independently drawn according to the conditional expectation $\mathbb{E}[y_j|h] = h, j \in [2r]$, or equivalently $\mathrm{Pr}[y_j = e_i|h] = h_i, j \in [2r], i \in [q]$.

Each observed variable $x_l$ for $l \in [2rn]$, is a discrete-valued $p$-dimensional random variable (word) where $p$ is the size of vocabulary. Again, we assume that variables $x_l$, are encoded by the basis vectors $e_k, \ k \in [p]$, such that $x_l = e_k \in \mathbb{R}^p$ when the $l$-th word in the document is $k$. Given the

corresponding topic $y_j, j \in [2r]$, words $x_l, l \in [2rn]$, are independently drawn according to the conditional expectation

$$\mathbb{E}\big[x_{(j-1)n+k}|y_j = e_i\big] = a_i, \ i \in [q], j \in [2r], \ k \in [n],$$

where vectors $a_i \in \mathbb{R}^p, \ i \in [q]$, are the conditional probability distribution vectors. The matrix $A = [a_1|a_2|\cdots|a_q] \in \mathbb{R}^{p \times q}$ collecting these vectors is called *population structure* or *topic-word matrix*.

The $(2rn)$-th order moment of observed variables $x_l, l \in [2rn]$, for some integer $r \geq 1$, is defined as (in the matrix form)

$$M_{2rn}(x) := \mathbb{E}\left[(x_1 \otimes x_2 \otimes \cdots \otimes x_{rn})(x_{rn+1} \otimes x_{rn+2} \otimes \cdots \otimes x_{2rn})^\top\right] \in \mathbb{R}^{p^{rn} \times p^{rn}}. \quad (1)$$

For the $n$-persistent topic model with $2rn$ number of observations (words) $x_l, l \in [2rn]$, the corresponding moment is denoted by $M_{2rn}^{(n)}(x)$.

In this paper, we consider the problem of identifiability when *exact* moments are available. Given $M_{2rn}^{(n)}(x)$, what are the sufficient conditions under which the population structure $A = [a_1|a_2|\cdots|a_q] \in \mathbb{R}^{p \times q}$ is identifiable? This is answered in Section 3.

## 3  Sufficient Conditions for Generic Identifiability

In this section, the identifiability result for the $n$-persistent topic model with access to $(2n)$-th order observed moment is provided. First, sufficient deterministic conditions on the population structure $A$ are provided for identifiability in Theorem 1. Next, the deterministic analysis is specialized to a random structured model in Theorem 2.

We now make the notion of identifiability precise. As defined in literature, (strict) identifiability means that the population structure $A$ can be uniquely recovered up to permutation and scaling for all $A \in \mathbb{R}^{p \times q}$. Instead, we consider a more relaxed notion of identifiability, known as generic identifiability.

**Definition 1** (Generic identifiability). *We refer to a matrix $A \in \mathbb{R}^{p \times q}$ as generic, with a fixed sparsity pattern when the nonzero entries of $A$ are drawn from a distribution which is absolutely continuous with respect to Lebesgue measure* [2]. *For a given sparsity pattern, the class of population structure matrices is said to be* generically identifiable [10], *if all the non-identifiable matrices form a set of Lebesgue measure zero.*

The $(2r)$-th order moment of hidden variables $h \in \mathbb{R}^q$, denoted by $M_{2r}(h)$, is defined as

$$M_{2r}(h) := \mathbb{E}\left[\left(\overbrace{h \otimes \cdots \otimes h}^{r \text{ times}}\right)\left(\overbrace{h \otimes \cdots \otimes h}^{r \text{ times}}\right)^\top\right] \in \mathbb{R}^{q^r \times q^r}. \quad (2)$$

**Condition 1** (Non-degeneracy). *The $(2r)$-th order moment of hidden variables $h \in \mathbb{R}^q$, defined in equation* (2)*, is full rank (non-degeneracy of hidden nodes).*

Note that there is no hope of distinguishing distinct hidden nodes without this non-degeneracy assumption. We do not impose any other assumption on hidden variables and can incorporate arbitrarily correlated topics.

Furthermore, we can only hope to identify the population structure $A$ up to scaling and permutation. Therefore, we can identify $A$ up to a canonical form defined as:

**Definition 2** (Canonical form). *Population structure $A$ is said to be in* canonical form *if all of its columns have unit norm.*

### 3.1  Deterministic Conditions for Generic Identifiability

Before providing the main result, a generalized notion of (perfect) matching for bipartite graphs is defined. We subsequently impose these conditions on the bipartite graph from topics to words which encodes the sparsity pattern of population structure $A$.

**Generalized matching for bipartite graphs:** A bipartite graph with two disjoint vertex sets $Y$ and $X$ and an edge set $E$ between them is denoted by $G(Y, X; E)$. Given the bi-adjacency matrix $A$, the notation $G(Y, X; A)$ is also used to denote a bipartite graph. Here, the rows and columns of matrix $A \in \mathbb{R}^{|X| \times |Y|}$ are respectively indexed by $X$ and $Y$ vertex sets. Furthermore, for any subset $S \subseteq Y$, the set of neighbors of vertices in $S$ with respect to $A$ is denoted by $N_A(S)$.

**Definition 3** ((Perfect) $n$-gram matching). *A $n$-gram matching $M$ for a bipartite graph $G(Y, X; E)$ is a subset of edges $M \subseteq E$ which satisfies the following conditions. First, for any $j \in Y$, we have $|N_M(j)| \leq n$. Second, for any $j_1, j_2 \in Y, j_1 \neq j_2$, we have $\min\{|N_M(j_1)|, |N_M(j_2)|\} > |N_M(j_1) \cap N_M(j_2)|$.*
*A perfect $n$-gram matching or $Y$-saturating $n$-gram matching for the bipartite graph $G(Y, X; E)$ is a $n$-gram matching $M$ in which each vertex in $Y$ is the end-point of exactly $n$ edges in $M$.*

In words, in a $n$-gram matching $M$, each vertex $j \in Y$ is at most the end-point of $n$ edges in $M$ and for any pair of vertices in $Y$ ($j_1, j_2 \in Y, j_1 \neq j_2$), there exists at least one non-common neighbor in set $X$ for each of them ($j_1$ and $j_2$).

Note that 1-gram matching is the same as regular matching for bipartite graphs.

**Remark 1** (Necessary size bound). *Consider a bipartite graph $G(Y, X; E)$ with $|Y| = q$ and $|X| = p$ which has a perfect $n$-gram matching. Note that there are $\binom{p}{n}$ $n$-combinations on $X$ side and each combination can at most have one neighbor (a node in $Y$ which is connected to all nodes in the combination) through the matching, and therefore we necessarily have $q \leq \binom{p}{n}$.*

**Identifiability conditions based on existence of perfect $n$-gram matching in topic-word graph:**
Now, we are ready to propose the identifiability conditions and result.

**Condition 2** (Perfect $n$-gram matching on $A$). *The bipartite graph $G(V_h, V_o; A)$ between hidden and observed variables, has a perfect $n$-gram matching.*

The above condition implies that the sparsity pattern of matrix $A$ is appropriately scattered in the mapping from hidden to observed variables to be identifiable. Intuitively, it means that every hidden node can be distinguished from another hidden node by its unique set of neighbors under the corresponding $n$-gram matching.

Furthermore, condition 2 is the key to be able to propose identifiability in the overcomplete regime. As stated in the size bound in Remark 1, for $n \geq 2$, the number of hidden variables can be more than the number of observed variables and we can still have perfect $n$-gram matching.

**Definition 4** (Kruskal rank, [11]). *The Kruskal rank or the krank of matrix $A$ is defined as the maximum number $k$ such that every subset of $k$ columns of $A$ is linearly independent.*

**Condition 3** (Krank condition on $A$). *The Kruskal rank of matrix $A$ satisfies the bound $\mathrm{krank}(A) \geq d_{\max}(A)^n$, where $d_{\max}(A)$ is the maximum node degree of any column of $A$.*

In the overcomplete regime, it is not possible for $A$ to be full column rank and $\mathrm{krank}(A) < |V_h| = q$. However, note that a large enough krank ensures that appropriate sized subsets of columns of $A$ are linearly independent. For instance, when $\mathrm{krank}(A) > 1$, any two columns cannot be collinear and the above condition rules out the collinear case for identifiability. In the above condition, we see that a larger krank can incorporate denser connections between topics and words.

The main identifiability result under a fixed graph structure is stated in the following theorem for $n \geq 2$, where $n$ is the topic persistence level.

**Theorem 1** (Generic identifiability under deterministic topic-word graph structure). *Let $M_{2rn}^{(n)}(x)$ in equation (1) be the $(2rn)$-th order observed moment of the $n$-persistent topic model, for some integer $r \geq 1$. If the model satisfies conditions 1, 2 and 3, then, for any $n \geq 2$, all the columns of population structure $A$ are generically identifiable from $M_{2rn}^{(n)}(x)$. Furthermore, the $(2r)$-th order moment of the hidden variables, denoted by $M_{2r}(h)$, is also generically identifiable.*

The theorem is proved in Appendix A of the long version [12]. It is seen that the population structure $A$ is identifiable, given any observed moment of order at least $2n$. Increasing the order of observed moment results in identifying higher order moments of the hidden variables.

The above theorem does not cover the case of $n = 1$. This is the usual bag-of-words admixture model. Identifiability of this model has been studied earlier [13], and we recall it below.

**Remark 2** (Bag-of-words admixture model, [13]). *Given $(2r)$-th order observed moments with $r \geq 1$, the structure of the popular bag-of-words admixture model and the $(2r)$-th order moment of*

*hidden variables are identifiable, when A is full column rank and the following expansion condition holds [13]*

$$|N_A(S)| \geq |S| + d_{\max}(A), \quad \forall S \subseteq V_h, \ |S| \geq 2. \tag{3}$$

*Our result for $n \geq 2$ in Theorem 1, provides identifiability in the overcomplete regime with weaker matching condition 2 and krank condition 3. The matching condition 2 is weaker than the above expansion condition which is based on the perfect matching and hence, does not allow overcomplete models. Furthermore, the above result for the bag-of-words admixture model requires full column rank of A which is more stringent than our krank condition 3.*

**Remark 3** (Recovery using $\ell_1$ optimization). *It turns out that our conditions for identifiability imply that the columns of the $n$-gram matrix[3] $A^{\odot n}$, are the sparsest vectors in $\mathrm{Col}\left(M_{2n}^{(n)}(x)\right)$, having a tensor rank of one. See Appendix A in the long version [12]. This implies recovery of the columns of A through exhaustive search, which is not efficient. Efficient $\ell_1$-based recovery algorithms have been analyzed in [13, 14] for the undercomplete case $(n = 1)$. They can be employed here for recovery from higher order moments as well. Exploiting additional structure present in $A^{\odot n}$, for $n > 1$, such as rank-1 test devices proposed in [15] are interesting avenues for future investigation.*

## 3.2   Analysis Under Random Topic-Word Graph Structures

In this section, we specialize the identifiability result to the random case. This result is based on more transparent conditions on the size and the degree of the random bipartite graph $G(V_h, V_o; A)$. We consider the random model where in the bipartite graph $G(V_h, V_o; A)$, each node $i \in V_h$ is randomly connected to $d_i$ different nodes in set $V_o$. Note that this is a heterogeneous degree model.

**Condition 4** (Size condition). *The random bipartite graph $G(V_h, V_o; A)$ with $|V_h| = q, |V_o| = p$, and $A \in \mathbb{R}^{p \times q}$, satisfies the size condition $q \leq \left(c\frac{p}{n}\right)^n$ for some constant $0 < c < 1$.*

This size condition is required to establish that the random bipartite graph has a perfect $n$-gram matching (and hence satisfies deterministic condition 2). It is shown that the necessary size constraint $q = O(p^n)$ stated in Remark 1, is achieved in the random case. Thus, the above constraint allows for the overcomplete regime, where $q \gg p$ for $n \geq 2$, and is tight.

**Condition 5** (Degree condition). *In the random bipartite graph $G(V_h, V_o; A)$ with $|V_h| = q, |V_o| = p$, and $A \in \mathbb{R}^{p \times q}$, the degree $d_i$ of nodes $i \in V_h$ satisfies the lower and upper bounds $d_{\min} \geq \max\{1 + \beta \log p, \alpha \log p\}$ for some constants $\beta > \frac{n-1}{\log 1/c}, \alpha > \max\left\{2n^2\left(\beta \log \frac{1}{c} + 1\right), 2\beta n\right\}$, and $d_{\max} \leq (cp)^{\frac{1}{n}}$.*

Intuitively, the lower bound on the degree is required to show that the corresponding bipartite graph $G(V_h, V_o; A)$ has sufficient number of random edges to ensure that it has perfect $n$-gram matching with high probability. The upper bound on the degree is mainly required to satisfy the krank condition 3, where $d_{\max}(A)^n \leq \mathrm{krank}(A)$.

It is important to see that, for $n \geq 2$, the above condition on degree covers a range of models from sparse to intermediate regimes and it is reasonable in a number of applications that each topic does not generate a very large number of words.

**Probability rate constants:** The probability rate of success in the following random identifiability result is specified by constants $\beta' > 0$ and $\gamma = \gamma_1 + \gamma_2 > 0$ as

$$\beta' = -\beta \log c - n + 1, \tag{4}$$

$$\gamma_1 = e^{n-1}\left(\frac{c}{n^{n-1}} + \frac{e^2}{1 - \delta_1} n^{\beta'+1}\right), \tag{5}$$

$$\gamma_2 = \frac{c^{n-1} e^2}{n^n (1 - \delta_2)}, \tag{6}$$

where $\delta_1$ and $\delta_2$ are some constants satisfying $e^2\left(\frac{p}{n}\right)^{-\beta \log 1/c} < \delta_1 < 1$ and $\frac{c^{n-1}e^2}{n^n} p^{-\beta'} < \delta_2 < 1$.

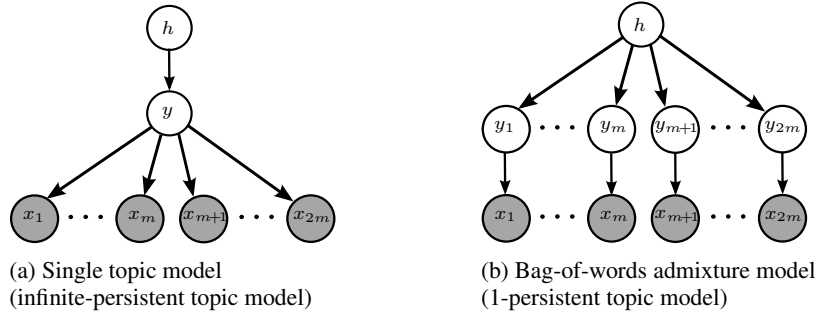

(a) Single topic model
(infinite-persistent topic model)

(b) Bag-of-words admixture model
(1-persistent topic model)

Figure 2: Hierarchical structure of the single topic model and bag-of-words admixture model shown for $2m$ number of words (views).

**Theorem 2** (Random identifiability). *Let $M_{2rn}^{(n)}(x)$ in equation (1) be the $(2rn)$-th order observed moment of the $n$-persistent topic model for some integer $r \geq 1$. If the model with random population structure $A$ satisfies conditions 1, 4 and 5, then **whp** (with probability at least $1 - \gamma p^{-\beta'}$ for constants $\beta' > 0$ and $\gamma > 0$, specified in (4)-(6)), for any $n \geq 2$, all the columns of population structure $A$ are identifiable from $M_{2rn}^{(n)}(x)$. Furthermore, the $(2r)$-th order moment of hidden variables, denoted by $M_{2r}(h)$, is also identifiable, **whp**.*

The theorem is proved in Appendix B of the long version [12]. Similar to the deterministic analysis, it is seen that the population structure $A$ is identifiable given any observed moment with order at least $2n$. Increasing the order of observed moment results in identifying higher order moments of the hidden variables.

The above identifiability theorem only covers for $n \geq 2$ and the $n = 1$ case is addressed in the following remark.

**Remark 4** (Bag-of-words admixture model). *The identifiability result for the random bag-of-words admixture model is comparable to the result in [14], which considers exact recovery of sparsely-used dictionaries. They assume that $Y = DX$ is given for some unknown arbitrary dictionary $D \in \mathbb{R}^{q \times q}$ and unknown random sparse coefficient matrix $X \in \mathbb{R}^{q \times p}$. They establish that if $D \in \mathbb{R}^{q \times q}$ is full rank and the random sparse coefficient matrix $X \in \mathbb{R}^{q \times p}$ follows the Bernoulli-subgaussian model with size constraint $p > Cq \log q$ and degree constraint $O(\log q) < \mathbb{E}[d] < O(q \log q)$, then the model is identifiable, whp. Comparing the size and degree constraints, our identifiability result for $n \geq 2$ requires more stringent upper bound on the degree ($d = O(p^{1/n})$), while more relaxed condition on the size ($q = O(p^n)$) which allows to identifiability in the overcomplete regime.*

**Remark 5** (The size condition is tight). *The size bound $q = O(p^n)$ in the above theorem achieves the necessary condition that $q \leq \binom{p}{n} = O(p^n)$ (see Remark 1), and is therefore tight. The sufficiency is argued in Theorem 3 of the long version [12], where we show that the matching condition 2 holds under the above size and degree conditions 4 and 5.*

## 4 Why Persistence Helps in Identifiability of Overcomplete Models?

In this section, we provide the moment characterization of the 2-persistent topic model. Then, we provide a discussion and comparison on why persistence helps in providing identifiability in the overcomplete regime. The moment characterization and detailed tensor analysis is provided in the long version [12].

The single topic model ($n \to \infty$) is shown in Figure 2a and the bag-of-words admixture model ($n = 1$) is shown in Figure 2b. In order to have a fair comparison among these different models, we fix the number of observed variables to $4$ (case $m = 2$) and vary the persistence level. Consider three different models: 2-persistent topic model, single topic model and bag-of-words admixture model (1-persistent topic model). From moment characterization results provided in the long version [12], we have the following moment forms for each of these models.

For the 2-persistent topic model with 4 words ($r = 1, n = 2$), we have

$$M_4^{(2)}(x) = (A \odot A)\mathbb{E}\big[hh^\top\big](A \odot A)^\top. \tag{7}$$

For the single topic model with 4 words, we have

$$M_4^{(\infty)}(x) = (A \odot A)\,\mathrm{Diag}\left(\mathbb{E}[h]\right)(A \odot A)^\top,\tag{8}$$

And for the bag-of-words-admixture model with 4 words ($r = 2, n = 1$), we have

$$M_4^{(1)}(x) = (A \otimes A)\mathbb{E}\left[(h \otimes h)(h \otimes h)^\top\right](A \otimes A)^\top.\tag{9}$$

Note that for the single topic model in (8), the Khatri-Rao product matrix $A \odot A \in \mathbb{R}^{p^2 \times q}$ has the same as the number of columns (i.e. the latent dimensionality) of the original matrix $A$, while the number of rows (i.e. the observed dimensionality) is increased. Thus, the Khatri-Rao product "expands" the effect of hidden variables to higher order observed variables, which is the key towards identifying overcomplete models. In other words, the original overcomplete representation becomes determined due to the 'expansion effect' of the Khatri-Rao product structure of the higher order observed moments.

On the other hand, in the bag-of-words admixture model in (9), this interesting 'expansion property' does not occur, and we have the Kronecker product $A \otimes A \in \mathbb{R}^{p^2 \times q^2}$, in place of the Khatri-Rao products. The Kronecker product operation increases both the number of the columns (i.e. latent dimensionality) and the number of rows (i.e. observed dimensionality), which implies that higher order moments do not help in identifying overcomplete models.

Finally, Contrasting equation (7) with (8) and (9), we find that the 2-persistent model retains the desirable property of possessing Khatri-Rao products, while being more general than the form for single topic model in (8). This key property enables us to establish identifiability of topic models with finite persistence levels.

**Remark 6** (Relationship to tensor decompositions)**.** *In the long version of this work [12], we establish that the tensor representation of our model is a special case of the Tucker representation, but more general than the symmetric CP representation [6]. Therefore, our identifiability results also imply uniqueness of a class of tensor decompositions with structured sparsity which is contained in the class of* Tucker *decompositions, but is more general than the* Candecomp/Parafac *(CP) decomposition.*

## 5   Proof sketch

The moment of $n$-persistent topic model with $2n$ words is derived as $M_{2n}^{(n)}(x) = (A^{\odot n})\,\mathbb{E}\left[hh^\top\right](A^{\odot n})^\top$; see [12]. When hidden variables are non-degenerate and $A^{\odot n}$ is full column rank, we have $\mathrm{Col}\left(M_{2n}^{(n)}(x)\right) = \mathrm{Col}\left(A^{\odot n}\right)$. Therefore, the problem of recovering $A$ from $M_{2n}^{(n)}(x)$ reduces to finding $A^{\odot n}$ in $\mathrm{Col}\left(A^{\odot n}\right)$. Then, under the *expansion condition* [4]

$$\left|N_{A_{\mathrm{Rest.}}^{\odot n}}(S)\right| \geq |S| + d_{\max}\left(A^{\odot n}\right), \quad \forall S \subseteq V_h,\ |S| > \mathrm{krank}(A),$$

we establish that, matrix $A$ is identifiable from $\mathrm{Col}\left(A^{\odot n}\right)$. This identifiability claim is proved by showing that the columns of $A^{\odot n}$ are the sparsest and rank-1 (in the tensor form) vectors in $\mathrm{Col}\left(A^{\odot n}\right)$ under the sufficient expansion and genericity conditions.

Then, it is established that, sufficient combinatorial conditions on matrix $A$ (conditions 2 and 3) ensure that the expansion and rank conditions on $A^{\odot n}$ are satisfied. This is shown by proving that the existence of perfect $n$-gram matching on $A$ results in the existence of perfect matching on $A^{\odot n}$. For further discussion on proof techniques, see the long version [12].

**Acknowledgments**

The authors acknowledge useful discussions with Sina Jafarpour, Adel Javanmard, Alex Dimakis, Moses Charikar, Sanjeev Arora, Ankur Moitra and Kamalika Chaudhuri. A. Anandkumar is supported in part by Microsoft Faculty Fellowship, NSF Career award CCF-1254106, NSF Award CCF-1219234, ARO Award W911NF-12-1-0404, and ARO YIP Award W911NF-13-1-0084. M. Janzamin is supported by NSF Award CCF-1219234, ARO Award W911NF-12-1-0404 and ARO YIP Award W911NF-13-1-0084.

## Footnotes

[2] As an equivalent definition, if the non-zero entries of an arbitrary sparse matrix are independently perturbed with noise drawn from a continuous distribution to generate $A$, then $A$ is called generic.

[3] $A^{\odot n} := \underbrace{A \odot \cdots \odot A}_{n \text{ times}}$.

[4] $A_{\mathrm{Rest.}}^{\odot n}$ is the restricted version of $n$-gram matrix $A^{\odot n}$, in which the redundant rows of $A^{\odot n}$ are removed.

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
