[Reviews · NeurIPS 2013]

Submitted by Assigned_Reviewer_4

This paper gives a sufficient condition for a unique recovery of topic-word matrix of "n-persistent" overcomplete topic models by a tensor decomposition of moments. In the overcomplete regime, the number of topics are more than that of words. Thus, it is impossible to uniquely recover the relation between topics and words. However, this paper overcomes this difficulty utilizing the notion "perfect n-gram matching". Utilizing this notion, sufficient conditions for identifiability are given for a wide range of situations. The result shows that as n, the length of persistency of a topic, gets larger, it becomes easier to recover the topic-word matrix uniquely.

The theoretical analysis given in this paper is quite novel. That can be applied to a wide range of situations (from sparse to dense models), and the conditions are intuitively meaningful.
The discussions about why persistence helps for identifiability is instructive.

My concern is about its readability. The intuition of some theoretical materials is sometimes unclear for non-experts. Moreover, some mathematical definitions are missing.

For example, it is hard for non-experts to catch the theoretical essence about why even-th order (2rn-th) moments are required and we need to split the 2rn words into 2 groups of rn words to obtain a matrix form of the moments. Some intuitive explanations about this point would be helpful.

Except the readability, this is an interesting paper.

==After the author feed-back
I appreciate the feed-back. Such explanations as given in the rebuttal would improve the readability, and as a result, the significance of the results could be highlighted more effectively.


Minor comments:
Khatri-Rao product is defined only in the supplementary material. Khatri-Rao product itself depends on a partitioning pattern of matrices. The partitioning pattern used in the analysis should be defined in the main body.
Summary: Unique recovery condition of the topic-word matrix of n-persistent topic models from moments is derived. More readable descriptions are welcome.

Submitted by Assigned_Reviewer_5

Identifiability of a model typically entails that the mapping from model parameters to the set of distributions is injective. In this paper they consider the mapping from model parameters to set of moments, and the class of topic (or probabilistic admixture models), and derive sufficient conditions for identifiability of such topic models. They do so by relating the identifiability of model parameters to the existence of a perfect "n-gram matching" in a bipartite graph. For the specific setting where each topic is randomly supported on a bounded subset of the set of words, they show that their identifiability condition is also necessary in part.

This is a well-written paper, with interesting results.

CONS:

The title of the paper suggests that they derive identifiability conditions for overcomplete latent variable models in general, but their technical development is very strongly tied to a topic model. Perhaps the title could be made more specific.
Summary: Well-written paper that provides identifiability conditions for topic models.

Submitted by Assigned_Reviewer_6

The paper is concerned with identifiability of latent variable topic models, characterizing when the latent structure is identifiable from observable moments in these models. The authors introduce a multi-view type of model called the n-persistent topic model where each set of n words (in sequence) share the same topic and study how n plays a role in identifiability.

The first theorem is a deterministic result providing sufficient conditions for identifying the population structure (or the topic distributions) from the (2rn)-th observable moment. The three conditions involve non-degeneracy of the prior distribution over topic probability, an non-overlapping condition on the sparsity pattern of the columns of the population structure matrix, and a condition relating the kruskal rank of the population matrix to the sparsity of each column.

The paper then analyzes population structures with random sparsity pattern and characterizes parameter settings (sparsity, degree of overcompleteness) under which these models are w.h.p. identifiable. Here, the authors show that if the number of topics is O(p^n) where p is the vocabulary size and n is the persistence parameter, and the support of each topic distribution is O(p^{1/n}), then these models are w.h.p. identifiable.

While the paper deals with matrix representations throughout, they comment that working with the tensor representations would yield uniqueness conditions for a range of tensor decompositions (depending on the persistence parameter). This is an interesting result which is presented as an afterthought in the paper but should be emphasized. Working with the tensor representations may also lend to readability.

The results of the paper are novel and interesting. The paper, however, suffers in terms of readability; some more intuition about the conditions and the main theorem would significantly improve the clarity of the paper. For example, condition 2 and 3 together imply that ideal matrices have high degree of sparsity while still containing a n-perfect matching but it's not obvious why these matrices are the ideal ones. Moreover, the awkward matrix notation makes the paper significantly harder to read.

Questions:
1. For condition 1 to be satisfied, it is essential that the topic proportions is a random quantity, but this is not always assumed in analysis for latent variable models (if h is fixed then M_{2r}(h) is always rank one). Why is this condition required? Previous analysis [9] seem to not require a prior distribution on the topic proportions while still guaranteeing identifiability.
Summary: The paper studies a problem that has received a lot of attention in the past few years and makes a nice contribution with interesting results and proof techniques. Deeper intuitions about the results could significantly improve the readability of the paper.
Author Feedback

Author rebuttal: Reviewer 4:

* Why split the 2rn words into 2 groups of rn words? Isn't it sufficient to consider just the rn words and their moments, E[x_1 \otimes x_2 ... \otimes x_{rn}], as in [9]?

Response: In our model, the decomposition of the tensor E[x_1 \otimes ...\otimes x_{2rn}] is overcomplete, so the techniques of [9] do not apply here. We use alternative techniques and for the purpose of analysis, we reshape the tensor E[x_1 \otimes ...\otimes x_{2rn}] into a matrix of rn by rn words, where the first group of words is {x_1,..,x_n} and the second group is {x_n+1,.. x_2n}. Note that each group is a sequence of n successive words and we assume that the topic persistence level is at least n. The order of the words is important when the persistence level n>1, since we assume that a common topic persists over a sequence of n words.

With regards to the question on why 2rn words are required and not just rn: Note that even for n=r=1, which is the case considered in [12], second order moment (2rn) are required and the first order moment (rn) is not sufficient. Similarly in [9], when n=r=1, we require third order moment and the first order moment is not sufficient. We establish that 2n-th moment is necessary for estimating overcomplete models when q=Theta(p^n) with persistence level n.

*Only have 2rn words and moments are estimated in a one-shot way. However, to estimate moments, need sufficient number of repetitions of words. Even if we have a sufficient number of words, there is still a problem about where we should split the words into groups of rn words. The problem setting (how to estimate the moments and construct samples) is seriously unclear.

Response: In a single document, 2rn is the minimum number of words required for estimating the moment. We estimate the moment tensor E[x_1 \otimes ...\otimes x_{2rn}] by considering the first 2rn words in each document and averaging over documents in the corpus to obtain a consistent estimate. Note that this is not a bag-of-words model (when persistence level>1), and the order of the words is important for estimating the moment. Once the tensor is computed, it can be appropriately reshaped into a matrix to obtain M_{2rn}(x).

**********Reviewer 5:

*title too broad

Response: Revised title is: When are Overcomplete Topic Models Identifiable? Uniqueness of Tensor Tucker Decompositions with Structured Sparsity

*****************Reviewer 6:

*Why not use tensor representation and not matrix form?

Response: We have provided the tensor form of the moment in the appendix, but since we felt that a part of NIPS audience may not be familiar with the tensor notation, we retained only matrix form in the main text. Our identifiability results imply uniqueness of a structured class of tensor Tucker decompositions and we will attempt to emphasize this in the main text in the revised version.

*More intuitions on degree and krank conditions 2 and 3

Response: The topic-word matrix must have sparsity ranging from sparse (log p) to intermediate regimes (p^{1/n}) for identifiability (in random setting). On one hand, too sparse models do not have enough edges in the bipartite graph from topics to words, and therefore, the different topics cannot be distinguished. On the other hand, if the bipartite graph is too dense, then there is not enough diversity in the word supports among different topics. Thus, we cannot have identifiability when the degrees are too small or too large, and we provide a range of degrees for identifiability Condition 2 ensures that this occurs through the requirement of a perfect n-gram matching. Regarding the relationship between krank and degrees in condition 3, intuitively, a larger krank leads to better distinguishability among the topics, and therefore, we can tolerate larger degrees when the krank is correspondingly larger. We will add this discussion to the revised version.

*For condition 1 to be satisfied, it is essential that the topic proportions is a random quantity. Previous analysis [9] seem to not require a prior distribution on the topic proportions while still guaranteeing identifiability.

Response: This is not true. [9] is restrictive and requires the topic proportions to be either drawn from Dirichlet distribution, or only have single topics in each document. Assuming these distributions implies that M_{2r}(h) is full rank. Our condition 1 is the natural non-degeneracy condition that each topic cannot be expressed as a combination of the other topics: if that were the case, we would not have identifiability. Thus condition 1 is necessary for identifiability. Condition 1 does not require a random model on h. All we require is that the topics be distinguishable from one another. Note that this condition is also required in [12], as well as in works of Arora et. al. for learning topic models via NMF techniques.